# Spatial index relating urban environment to health lifestyle and obesity risk in men and women from different age groups

**David Michel Oliveira**[1], **Mara Lucia Marques**[2], **Daniel dos Santos**[3], **Maria Claudia Bernardes Spexoto**[4‡], **Giovanna Benjamin Togashi**[5‡], **Danilo Alexandre Massini**[6,7], **Dalton Müller Pessôa Filho**[6,7¤‡*]

**1** Federal University at Goiás (UFG/UFJ), Jataí, Goiás, Brazil, **2** Pontifical Catholic University of Campinas (PUC-Campinas), CEATEC, Faculty of Geography, São Paulo, Brazil, **3** Postgraduate Program in Health Promotion, University of Franca (UNIFRAN), Franca, São Paulo Brazil, **4** Federal University at Grande Dourados (UFGD), Dourados, Mato Grosso do Sul, Brasil, **5** Sesc São Paulo, São Paulo, Brazil, **6** Postgraduate Program in Human Development and Technology, Institute of Biosciences, São Paulo State University (UNESP), Rio Claro, São Paulo, Brazil, **7** Department of Physical Education, College of Sciences, São Paulo State University (UNESP), Bauru, São Paulo, Brazil

☯ These authors contributed equally to this work.
¤ Current address: Department of Physical Education, São Paulo State University (UNESP), Bauru, São Paulo, Brazil
‡ These authors also contributed equally to this work.
* dalton.pessoa-filho@unesp.br

**Data Availability Statement:** All relevant data are within the manuscript and its Supporting Information files.

## Abstract

The challenge in the search for relationships between urban space, physical mobility, and health status, is detecting indicators able to link the environment with healthy life habits. Therefore, the objective was to design an urban index for the identification of urban environment propensity for physical activity (PA) and to determine how it relates to lifestyle and anthropometric parametrization of obesity. Participants (N = 318—60.4% women and 39.6% men) were recruited from a mid-sized city with epidemiology and morbidity rates below the average for the mid-west region of Brazil. Body mass index (BMI) was measured and a questionnaire was applied to gather information about PA and life habits. The spatial urban health index (SUHI) was designed in a geographic information system using data from demographic, environmental and urban physical features. The relationship between BMI and PA was verified with multiple linear regression, controlled for SUHI levels. Regarding the BMI of the population, 69.5% were classified in the eutrophic or overweight ranges, with no effect of gender and age. The SUHI classified 63.7% of the urban area favorable to PA. The PA routine was adequate ($\geq$3 sessions with $\geq$1 h each) for ~80% of the population, as well as healthy habits such as non smoking (~94%) and non alcohol abuse (~55%). The SUHI strengthens the relationships of BMI to weekly frequency (r = -0.68; t = -9.4; p<0.001) and session duration (r = -0.66; t = -2.8; p<0.001) for the whole group by improving the explanatory coefficient in ~25% ($R^2_{Adj}$ = 0.61 to $R^2_{Adj}$ = 0.85). The SUHI indicated that the urban environment is able to promote healthy life habits by diminishing the "obesogenic" features of the city when physical structures are planned to facilitate PA, whatever the gender and age group.

**Funding:** The author(s) received no specific funding for this work.

**Competing interests:** The authors have declared that no competing interests exist.

## Introduction

The incentive to practice physical activities in public environments is a trend in health policy strategies, aiming at reducing the propensity for chronic noncommunicable diseases and promoting healthy habits by increasing physical activity (PA) among the population [1–2].

Although socioeconomic aspects are believed to be associated with lifestyle and health perception [2–3], there are actions directly derived from health policies and associated with economic and social policies which are intended to guarantee physical, mental, and social well-being conditions, such as measures to reduce the risk of cardiovascular diseases, metabolic disorders, and obesity and thus promote collective and individual health [4–5]. Among these measures, access to healthy environments, products, and services stand out [6]. Urban indicators include characteristics of the constructed environment (e.g. neighborhood organization, building density, type of use, and quality of the housing) and access to public services (e.g. transport services, green areas, quality and security of public spaces), which make the urban area more efficient regarding energy expenditure, favoring more active mobility and promoting health benefits [6–7]. This approach has been called the ecosystem approach to human health, which is based on the integration between health of the population, management strategies, and the environment [8].

One of the challenges of this type of approach is to detect indicators that combine environmental features and health [5,8]. The inclusion of the environment as an interdisciplinary tool in the fight against obesity is based on the fact that PA in urban spaces is an important component in the strategy to prevent risk of mortality from metabolic and functional diseases, for both men and women in different age groups [4,7]. This association was observed by Andersen et al. [9], who found a 40% reduction in the risk of mortality among bicycle riders in cycling to work and carrying out leisure activities in Copenhagen, Denmark. Other studies have failed to show direct relationships, but do not disprove the beneficial effect of the urban environment on the reduced prevalence of obesity and overweight, unhealthy habits (smoking and physical inactivity), diabetes mellitus, self-deprecation, and risk of mortality from cardiovascular and respiratory diseases [2,10–12]. The most consistent results on the relationship between environment and health highlight the stimulating effect of green areas on PA, which supposedly modulates the beneficial action on obesity rates and risk of cardiovascular diseases, but there is no consensus on the joint action of green spaces, PA, and obesity [13]. It is also noteworthy that environmental and population factors that influence this relationship are little explored, such as other characteristics of space beyond green areas (e.g. population density and access roads) and the weighting of the relationships by socioeconomic factors, gender, and age [13].

The objective of the present study was to develop an urban index to identify the city's propensity to practice PA, intending to relate it to obesity anthropometric indicators and exercise structure parameters, as well as to contextualize the life habits of the analyzed population. The justification for proposing obesity as a study object is supported by the fact that, despite the growing scientific concern about the different factors related to overweight, materialized as measures that range from biomedical interventions for its treatment to public prevention policies that include environmental aspects, the cases of obesity still show alarming data [1,2,5,14–15].

The intention was to provide information on the interaction between characteristics of the urban space capable of conditioning the combination between PA and obesity indicators as the primary outcome. Additionally, the study aimed at evaluating life habits, which are a confounding variable because they enhance the effect of PA on obesity control [12,15]. The present study included an "on-site" analysis of a population that uses urban spaces for PA, which is an uncommon condition of analysis because of the scope of distance surveys, but

incorporating the verification of evidence to support the environment as a mediator of the relationship between PA and health indicators for a population sample [7,11–12].

## Material and methods

### Target population

The sample was 318 people of both genders (60.4% women and 39.6% men) who practiced PA. The inclusion criteria were: people should be 18 years of age or older and practice PA without supervision. The exclusion criteria were: pregnant women, people with special needs or physical limitations, and non-literate people. All procedures were clarified to the participants, who then signed a free and informed consent form. The study proposal was approved by the research ethics committee of the Federal University of Goiás (UFG-GO) (UFG: 1.641.233) and followed the ethical principles of the Declaration of Helsinki [16].

### Geographic contextualization of the study location

The municipality of Jataí is located in the southwest region of the state of Goiás, Brazil, has an area of 7,174.225 km$^2$, and a population of 88,006 people according to the 2010 population census, but with an estimated population in 2017 of 98,128 people [17]. The municipality had a per capita income of R$ 40,023.17 in 2015 and a municipal human development index of 0.757 [18]. Its relief is flat and mildly undulating, with about 95% of the area classified as a flat surface region and an altimetric variation between 500 and 1,100 m.

### Procedures and instruments

Field data collection was carried out in urban public environments that showed the highest attendance rates of PA practitioners and with suitable structures such as walking/running tracks and outdoor gyms. The procedures were applied considering the times with the highest attendance rates, being between 6 and 9 am and from 5 to 7 pm in October and November 2016 and January to June 2017.

**Anthropometric indicators.** Body mass (kg) and height (cm) were collected to calculate the body mass index (BMI). A digital scale (Wiso®) with a tare of 150 kg was used to obtain the body mass and a portable stadiometer (Sanny®) was applied to measure height. To measure the abdominal circumference (AbC), an inelastic tape measure (Sanny®) was used. For women, a reference value for AbC ≥ 88 cm indicates risk of metabolic complications, and for men this reference value is AbC ≥ 102 cm. For BMI (kg/m$^2$), the following classifications were adopted for both genders: normal: 18.5 to 24.9; overweight: 25.0 to 29.9; obesity: 30.0 to 34.9; severe obesity: ≥ 35. All the measurements performed and classifications used followed the WHO recommendations [19], with the dataset from the present study available in the S1 Table. Basically, WHO [19] suggest the cross-hand technique use for measuring abdominal circumference (AbC). The reading was taken from point zero of the tape measure by an evaluator located at the side of the participant. The tape should be at right angles to the reference point on the segment being measured, and the tension of the tape must be constant. The juxtaposition of the tape was required to ensure continuity of the points and it is recommended to observe the measurement at the level of the tape to avoid error. The AbC measured the perimeter at the narrowest point between the lower costal (10th rib) border and the iliac crest. The participant should not hold their breath during the procedure.

**Life habits, health data, and habitual physical activity.** A semi-structured questionnaire was applied through face-to-face interviews. The questionnaire consisted of: a) identification data: gender and age; b) health-related behaviors: alcohol consumption; smoking and diet; and

c) profile and practice of PA in ecological parks: time spent in public spaces, modality(ies) practiced, weekly frequency, session duration (hours/minutes), objective or motivation to perform PA, and importance of the presence of physical education professionals in public parks. The questionnaire applied in the present study is available in the S2 Board.

Incidence and death rates related to infarction, obesity, and diabetes mellitus were obtained by consulting DATASUS [20] for the municipality of Jataí, the state capital Goiânia, and the state of Goiás. Regarding public health indicators, the population of Jataí shows mortality coefficients (per 100,000 inhabitants) for acute myocardial infarction equal to 11.7, which is lower than the coefficients for Goiânia (24.6) and the state of Goiás (29.2). The municipality of Jataí also has a lower diabetes mellitus coefficient (14.0) in comparison to those obtained for Goiânia (20.5) and Goiás (19.7). Comparison of the percentages of hospital admissions resulting from metabolic and nutritional endocrine diseases, and diseases of the circulatory system considering all age groups showed that Jataí also has lower or similar rates (2.1% and 11.5%) in comparison to those found for Goiânia (4.0% and 11.3%) and Goiás (4.0% and 11.0%). The health reports are available in S3–S5 Tables.

**Spatial urban health indicators.** To identify the spatial urban health index (SUHI), data on demographic, environmental, and urban structure were collected. The number of residents in the enumeration area was taken from the 2010 demographic census [18]. Parameters regarding road connectivity and land use and vegetation cover were obtained from the Goiás State Information System (SIEG) [21]. From the land use and vegetation cover map, the areas of natural vegetation in the urban area were selected and used to estimate green areas per inhabitant and reclassified according to the index proposed by the WHO [22]. The density of access roads was expressed as the length of roads per $km^2$. The urban area slope parameter was generated by the Shuttle Radar Topography Mission (MDE–SRTM), with a spatial resolution of 30 m. Table 1 describes the parameters and indicators used in the definition of the SUHI (Eq 1), as well as the weighting of these indicators ranging from 1 to 10. Further information detailing the procedures step-by-step for the cartographic modelling of SUHI can be find at protocols.io under de registration number (dx.doi.org/10.17504/protocols.io.bb3jiqkn).

$$SUHI = D_{populational} + D_{green\ area} + Slope + D_{roads\ connection} \tag{1}$$

## Statistical treatment

The BMI and AbC indicators were treated for normality by applying the Shapiro-Wilk test. The values of BMI and AbC were compared to verify the effect of gender and age by applying ANOVA (two-way, and Bonferroni as a post hoc test), thus if difference is observed it can distinguish the risk of obesity according to gender and age. To explore how the risk of obesity is related to the parameter of cardiovascular risk and physical activity schedule and to verify the role of SUHI on such relationship, multiple linear regression using the least squares method correlated the BMI parameter (as a dependent factor) with AbC, weekly frequency, and session duration (as independent factors), considering the effect of gender and age, together and separately, weighted or not by the SUHI. To assure the confidence to the correlation values, measurements of variability and dispersion were calculated using Pearson's coefficient (r), sample-adjusted coefficient of variance ($R^2_{Adj}$), and standard error of the estimate (SEE). The significance level was adjusted to $p \leq 0.05$. To obtain the spatial primary products, a geographic database was prepared with SIG ArcGis 10.5.1 (ESRI v. 2018) for the composition of the physical and demographic analysis parameters (Table 1) and mapping of the SUHI by using the weighted multi-criteria overlay method (Eq 1), which allowed to obtain a representativeness index ($km^2$), classified by applying the quantile method, into four gradation levels ranging

**Table 1. Criteria for urban pattern classification.**

| Parameters | Indicators | Variables | Rate |
|---|---|---|---|
| Number of people per sector | Population density (people/km$^2$) $D_{population}$ | 4–300 | 1 |
| | | 300–1300 | 1 |
| | | 1300–2500 | 3 |
| | | 2500–3500 | 4 |
| | | 3500–4500 | 5 |
| | | 4500–5500 | 7 |
| | | 5500–8500 | 8 |
| | | 8500–14197 | 10 |
| Urban green space per people | Urban green space density (green m$^2$/people) $D_{green\ area}$ | < 12 | 1 |
| | | 12–20 | 5 |
| | | 20–36 | 8 |
| | | > 36 | 10 |
| Relief | Surface slope (%) *Slope* | 0–3 (very gently) | 10 |
| | | 3–8 (gently) | 9 |
| | | 8–20 (moderate) | 5 |
| | | 20–45 (steep) | 2 |
| | | > 45 (very steep) | 1 |
| Connectivity roads | Road density (road km/km$^2$) $D_{road\ connection}$ | < 5 | 1 |
| | | 5–15 | 5 |
| | | 15–25 | 7 |
| | | 25–35 | 9 |
| | | 35–45 | 10 |

from unfavorable to very favorable. Descriptive statistics and exploration of normality were performed using SPSS (v.18, Inc., USA), with the entire dataset available in the S6 Table.

The sampling power for associations between the dependent (BMI) and independent variables was determined considering the sample size (= 318). The input parameters were: (a) Pearson's coefficient "r"; (b) $Z\alpha = 1.96$ for an index $\alpha = 0.05$; and (c) assuming a safety level of $\beta = 1.282$ for a sample with a minimum power of 80% ($\beta = 0.20$), as follows:

$$Z_{1-\beta} = \sqrt{n-3}\frac{1}{2}Ln\left(\frac{1+r}{1-r}\right) - Z_{1-\alpha/2} \tag{2}$$

Magnitude-based inferences analysis was also applied to test the chances of the true magnitude of an effect. The probabilities were assessed qualitatively using the following scale: < 1% = extremely unlikely; 1% to 5% = very unlikely; >5% to 25% = unlikely; >25% to 75% = possibly; >75% to 95% = likely; >95% to 99.5% = very likely; and > 99.5% = extremely likely. This procedure ensures that when repeating the study several times, the sampling distribution of z = 0.5 ln ((1 + r) / (1—r)) will tend approximately to normality with a variance equal to 1/(n-3) [23]. Taking sample power and magnitude-based inferences together, it is possible to analyze how robust the information is regarding its ability to ensure the presence of the correlation between indices of obesity and exercise practice parameters for the actual population, and how prudent it is to infer if this effect exists in another population.

## Results

Table 2 shows the anthropometric indexing by BMI for men and women distributed by age groups. Regarding the BMI of the population, 69.5% were classified as eutrophic or overweight

**Table 2. Body mass index (BMI) and Abdominal Circumference (AbC) average values and risk classification for practitioners of Physical Education (PA).** Sample stratified by age and gender: women (N = 192) e men (N = 126).

| Gender | Indexes | | Age (years) | Population (%) |
|---|---|---|---|---|
| | BMI (kg/m$^2$)[a] | AbC (cm)[b] | | |
| Women | 27.2 ± 5.1 | 89.4 ± 13.6*† | 22.9 ± 3.4 | 19.8 |
| | 28.6 ± 5.4 | 93.1 ± 11.8 | 33.9 ± 2.7 | 34.1 |
| | 26.8 ± 4.6 | 89.3 ± 11.9 | 44.7 ± 2.6 | 36.5 |
| | 29.9 ± 4.9 | 97.3 ± 12.9* | 53.7 ± 3.1 | 32.5 |
| | 26.9 ± 4.9 | 95.2 ± 12.1*† | 66.1 ± 6.3 | 29.4 |
| Men | 26.7 ± 6.2 | 91.8 ± 20.0*† | 23.6 ± 3.5 | 10.9 |
| | 28.8 ± 4.3 | 99.6 ± 11.2 | 35.7 ± 2.8 | 9.9 |
| | 29.3 ± 3.9 | 99.7 ± 13.9 | 45.4 ± 2.6 | 12.0 |
| | 28.2 ± 2.7 | 99.9 ± 6.1* | 54.4 ± 2.9 | 14.6 |
| | 27.6 ± 2.5 | 102.2 ± 10.6*† | 69.3 ± 6.1 | 18.2 |
| **Indexes** | **Gender** | | **Cardiovascular Risk** | **Population (%)** |
| AbC (cm) | Women | Men | | |
| | < 80 | < 90 | absent | 16 % and 24 % |
| | ≥ 80 | ≥ 90 | present | 84 % and 76 % |
| BMI (kg/m$^2$) | **Both genders** | | **Classifcation** | **Population (%)** |
| | 18.5–24.9 | | Normal | 27.0 % |
| | 25.0–29.9 | | Overweight | 42.8 % |
| | 30.0–34.9 | | Obesity | 23.0 % |
| | ≥35.0 | | Severe Obesity | 7.2 % |

Obs.

[a]BMI (Body Mass Index)

[b]AbC (Abdominal Circumference). No difference was observed at $p \leq 0.05$ for BMI between genders and age ranges. The AbC values showed differences between age ranges according to *$p \leq 0.05$ and †$p \leq 0.01$, but not between genders.

(Table 2), without taking into account the effect of gender and age on BMI values (gender: F = 0.954, p = 0.33; age: F = 0.717, p = 0.58). Regarding AbC, the male population of young people between 18 and 29 years old showed desirable values (Table 2), which differed for both genders to those obtained for the 50 to 59 years (p = 0.04) and >60 years (p<0.01) age groups. The sample between 30 and 59 years old showed borderline high values, while for the elderly (≥ 60 years) the values were above the desirable limit (Table 2). For the >60 years group, AbC values also differed from those for the 30 to 39 years (p = 0.02) and 40 to 49 years (p = 0.01) groups. Among women, AbC had average values above 88 cm, ranking 84% of this population within the risk zone for cardiovascular disease, with a higher risk for women aged 50 to 60 years (Table 2). Among men, AbC measures classified as ≥90 cm accounted for 76% of the population (Table 2) for all the age groups.

Regarding the lifestyle of the examined population, Table 3 shows the reports of the analyzed men and women by percentage. Overall, 65.4% reported that they did not smoke, 55.3% reported that they did not consume alcohol, and 62.3% reported that they did not follow a dietary plan. Concerning the period of adherence to PA in urban spaces, the majority (71%) had been engaged for over a year. When asked about the preferred exercise modality, walking accounted for 96% of the answers. Exploring aspects of the PA routine revealed that 87% practiced it more than three times per week and 73% reported that their exercise sessions lasted one hour or more. Participants were almost unanimous (95%) in answering about the importance of the presence of a professional to guide the exercises at the times during which the number of practitioners at the facilities is higher.

**Table 3. Life habits and physical activity (PA) routine.** Sample stratified by age and gender: women (N = 192) and men (N = 126).

| Parameters for PA | Total (%) | Women (%) | Men (%) |
|---|---|---|---|
| Modality | | | |
| Walking | 95.9 | 97.9 | 92.9 |
| Running | 4.1 | 2.1 | 7.1 |
| Place | | | |
| Gym | 0 | 0.0 | 0.0 |
| Public spaces | 100 | 100.0 | 100.0 |
| Motivation | | | |
| Health | 87.4 | 87.5 | 87.3 |
| Aesthetic | 6.6 | 9.4 | 2.4 |
| Sociabilization | 0.3 | 0.0 | 0.8 |
| Stress reduction | 2.2 | 2.1 | 2.4 |
| Leisure | 3.5 | 1.0 | 7.1 |
| Frequency (section per week) | | | |
| 1/week | 18.3 | 20.0 | 15.1 |
| 2/week | 6.6 | 7.8 | 4.8 |
| 3/week | 20.8 | 18.8 | 23.8 |
| > 3/week | 42.1 | 42.7 | 41.3 |
| Every day of the week | 12.3 | 10.4 | 15.1 |
| Regularity | | | |
| First time | 2.5 | 3.1 | 1.6 |
| One week | 6.3 | 6.8 | 5.6 |
| One month | 3.5 | 3.6 | 3.2 |
| 1–3 months | 6.9 | 6.3 | 7.9 |
| 3–6 months | 2.8 | 2.6 | 3.2 |
| 6–9 months | 5.3 | 5.2 | 5.6 |
| 9–12 months | 2.5 | 1.0 | 4.8 |
| More than 12 months | 70.1 | 71.4 | 68.3 |
| Duration (per section) | | | |
| 60–120 minutes | 73.3 | 71.9 | 75.4 |
| <60 minutes | 22.3 | 24.0 | 19.8 |
| >120 minutes | 4.4 | 4.2 | 4.8 |

Obs.: "%" percentage of people engaged in whole sample and by gender.

The adopted criteria, listed in Table 1, allowed the elaboration of a spatial analysis, aiming at obtaining a physical and demographic classification to indicate whether the environment favors the practice of PA (Fig 1). Considering the population density, green area, slope, and density of access roads, the following urban area SUHI was obtained: 7.24% (~3,634 km$^2$) were unfavorable, 29.09% (~14,604 km$^2$) were a somewhat favorable, 42.54% (~21,357 km$^2$) were favorable, and 21.14% (~10,613 km$^2$) were strongly favorable to PA (Fig 1). The variation in the BMI (27.8 ± 4.6 kg/m$^2$) in the total population was related to changes in AbC (94.4 ± 12.8 cm) and the weekly frequency (3.70 ± 0.98 sessions) and duration (67.4 ± 29.6 minutes) of the PA sessions, with potential (r = 0.78; R$^2_{Adj}$ = 0.61; p<0.001; and SEE = 2.9 kg/m$^2$), high sample power (>99.9%), and an extremely likely (100%) probability of occurrence in other populations. Considering the weight of the SUHI rate (favorable area/unfavorable area = 7.4 km$^2$ on average), there was an increase in the explanatory potential of independent over dependent

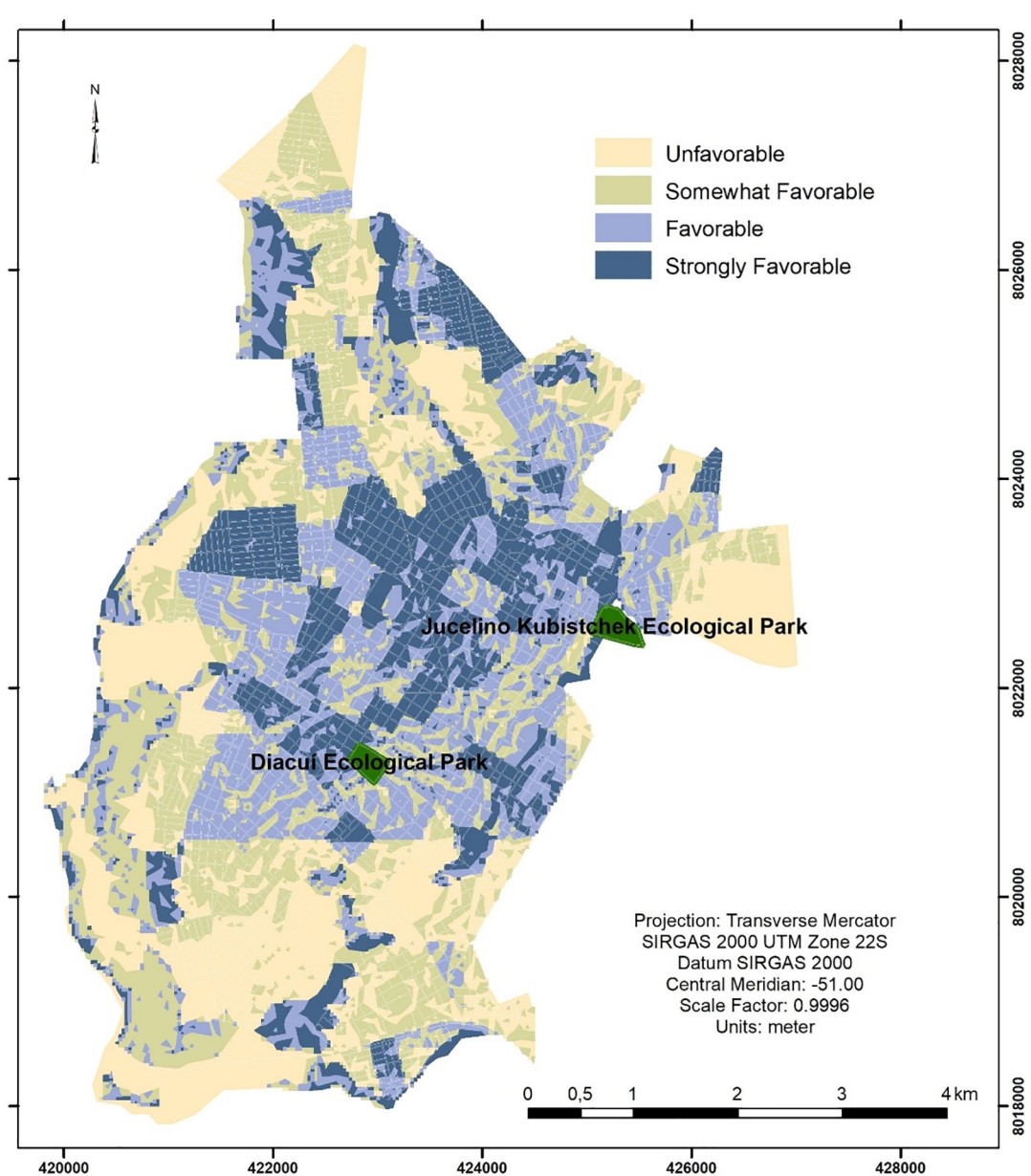

**Fig 1. Classification of urban environment ability to promote physical activity according to spatial and demographical characteristics.** Cartographical output produced by the authors, with original content modelled from open-access information not requiring license, permission and copyright declaration.

parameters (r = 0.92; $R^2_{Adj}$ = 0.85; p<0.001), showing direct relationships between BMI and AbC (r = 0.88; t = 24.9; p<0.001) and indirect relationships with weekly frequency (r = -0.68; t = -9.4; p<0.001) and session duration (r = -0.66; t = -2.8; p<0.01).

However, this combination of effects on BMI (weighted by the SUHI) existed only among men (AbC: r = 0.73; t = 9.9; p<0.001; weekly frequency: r = -0.78; t = -7.7; p<0.001; and session duration: r = -0.69; t = - 4.9; p<0.001), especially those in the 50 to 59 (AbC: r = 0.82; t = 13.0; p<0.01; weekly frequency: r = -0.294; t = -2.5; p = 0.02; and session duration: r = -0.34; t = -2.9; p<0.01) and the >60 years age groups (AbC: r = 0.76; t = 6.5; p<0.001; weekly frequency: r = -0.81; t = - 6.6; p<0.001; and session duration: r = - 0.77; t = - 2.9; p<0.01). In

all cases, the sampling power ranged from 97.1% to 99.9% with a very high probability (100%) of the SUHI rate being applied to the urban population of other cities. However, among women, only the AbC (r = 0.87; t = 24.5; p<0.001) had an effect on BMI when compared to the SUHI weighting.

## Discussion

Regarding anthropometric health indicators, AbC values indicated increased cardiovascular risk for men in all age groups. The BMI also classified the total population (men and women in all the age groups) as having excess body weight (overweight). These results align with the negative aspects related to inadequate levels of PA and advanced age, such as reduced lean tissue and increased fat tissue, in a continuous cycle that elevates the risk of cardiovascular diseases and obesity combined with a sedentary lifestyle and aging [15,24], although most of the analyzed population reported regular practice of PA, with weekly frequency (four to five sessions per week) and duration (one hour of PA per session) consistent with recommendations, and stated that they did not smoke and/or consume excess alcohol. The absence of dietary control (reported by 86.2% of the examined men and 58% of the studied women) and professional orientation for ideal PA may explain the high AbC and BMI values in this population [24–26].

Spatial analysis revealed that mobility on foot and PA are favored by the interaction of urban elements such as population density, green area, slope, and density of access roads, which combined to characterize approximately 63.7% of the urban area as conducive to PA. The results of the present study also indicate that the aspects of the urban environment indexed by the SUHI make the practice of PA negatively related to the obesity measurement (BMI). However, this result was observed only among middle-aged (50 to 59 years old) and elderly (> 60 years old) men, which is probably due to the fact that exercises are performed more vigorously by the male population, as pointed out by Cohen et al. [27]. Other factors not explored by the present study have been speculated as being capable of impacting on the phenomenon, given that they independently affect men and women regarding the obesogenic influence of the environment. Some of these factors include physical (structure and facilities), economic (income and food trade), political (safety and usage guidelines), and sociocultural (family and community models) aspects of the local environment [5,28–29], which may differ in small and medium-sized cities when compared to large cities (large cities being predominantly analyzed) [3,5,7].

The direct observations carried out in the present study indicate that the examined PA was performed with a minimum compliance to current recommendations, but there was imperfection when the time of practice and type of exercise performed are considered, which culminate in neglecting the progression aspect, especially in the walking option, which was the preferred type of exercise, and in the lack of professional supervision, which is inherent in the success of long-term planning [25–26]. While walking facilitates engagement to perform PA in overweight (BMI > 26) and obese (BMI > 30) people, it may not meet the recommendations of exercise intensity aiming at obtaining a pronounced effect on fat and weight, mainly among people engaged in long-term PA [24,26]. In addition, there is the effect of weekly load (planning), which influences the effectiveness of body weight loss associated with continuous PA at a moderate intensity, over medium (> six months) and long (> 12 months) periods of time [14,25]. The absence of a direct correlation between the SUHI with PA practice and obesity indicators in the examined population may be in line with the imperfect planning of PA practice. Consequently, the present study suggests that the role of the urban environment is to favor PA and strengthen the relationship between exercise and reduction of excess body weight, as observed by Charreire et al. [30] in finding that regions with a high density of green areas and ease of access encourage the practice of exercise but have no effect on BMI.

Among the analyzed population, there is not only an awareness that exercise is a component of healthy habits, but also an encouragement to do so, associated with a lower prevalence of smoking and alcohol consumption, which tends to make the PA level minimally sufficient to avoid metabolic disorders and excess body weight [14,24]. Therefore, it is important to emphasize that a favorable urban environment, as indexed by the SUHI, is an incentive to adopt healthy habits that are fundamental for reducing or controlling obesity, regardless of the size of its area, a fact that has been reported only for large cities and indexed mainly by green areas or socioeconomic attractions [28,30].

Finally, it is noteworthy that the absence of a reference for the intensity of the physical effort during PA and for the evaluation of the physical fitness of the population limits the comprehension of the impact of the SUHI on physical mobility and exercise effectiveness for the purpose of reducing body weight. However, this limitation is common to all studies on the subject and remains to be investigated. In addition, as the population analyzed in the present study included physical activity practitioners in the majority, it should be highly recommended to include a sample of non-physical activity practitioners in future studies. This will, enable the analysis of whether spatial urban features characterized by SUHI also strengthens (or not) the correlation (if any) to the parameters of obesity risk among those visiting urban public spaces (or parks) occasionally (not necessary for the practice of physical activity), or have never used such spaces for their living activities. Moreover, the use of tools to improve the robustness of obesity assessment (bio-impedance or dual X-ray absorptiometry), as well as clinical tests for the diagnostics of diabetes and cardiovascular diseases will certainly provide in future studies unrefuted evidence for the role or spatial urban spaces to stimulate healthy life habits and reduced risks of obesity and comorbidities.

## Conclusion

Practitioners of PA in urban environments showed anthropometric indicators characteristic of overweight and obesity, but had Brazilian Ministry of Health epidemiology and morbidity indices lower than the average values obtained for the region, suggesting that the reported healthy life habits (practicing regular PA and avoiding smoking and excessive alcohol consumption), with the SUHI weighting to correlate negatively with BMI, is a reasonable inference to an integrative effect between urban space, life habits and physical activity able to reduce sedentary and obesogenic tendencies, even though the exercise plans originated caveats regarding the effectiveness of obesity control by accommodating the intensity-volume ratio. Therefore, it is evident that urban spatial characteristics are decisive to achieve this healthy behavior because they strengthen the correlation between PA and obesity indicators. In the present study, this urban propensity for mobility was quantified using the SUHI, which is the contribution to future studies that aim at analyzing the role of different physical elements of the cities as health promoters. From the results, it is recommended that the existence of a direct causal relationship between the SUHI and the physical condition of the urban population be examined and that urban policies aim to expand the areas favorable to PA in the urban area, as a promising social bias in promoting less obesogenic environments. Therefore, the SUHI provides relevant understanding of how the features of urban environment affect healthy behaviors whatever the gender and age group, but specially middle age and older men.

## Supporting information

**S1 Table. Excel file for BMI and AbC classification.**
(XLSX)

**S2 Table. Board Word file with questionnaire in both English and Portuguese languages.**
(DOCX)

**S3 Table. Excel file with health reports for Jataí.**
(XLS)

**S4 Table. Excel file with health reports for Goiania city.**
(XLS)

**S5 Table. Excel file with health reports for Goiás State.**
(XLS)

**S6 Table. SPSS file with dataset for treatment.**
(SAV)

## Acknowledgments

We extend special thanks to all study participants and to the technical support of the students (from Research Group "Physical Exercise and Nutrition applied to Health Promotion and Human Performance") during testing and recruitment. We also thank PUC-Campinas for their collaboration on the use of ArcGIS software.

## Author Contributions

**Conceptualization:** David Michel Oliveira, Daniel dos Santos, Giovanna Benjamin Togashi, Dalton Müller Pessôa Filho.

**Data curation:** Mara Lucia Marques, Dalton Müller Pessôa Filho.

**Formal analysis:** David Michel Oliveira, Mara Lucia Marques, Daniel dos Santos, Maria Claudia Bernardes Spexoto, Giovanna Benjamin Togashi, Danilo Alexandre Massini, Dalton Müller Pessôa Filho.

**Funding acquisition:** David Michel Oliveira, Daniel dos Santos.

**Investigation:** David Michel Oliveira, Mara Lucia Marques, Daniel dos Santos, Maria Claudia Bernardes Spexoto, Giovanna Benjamin Togashi, Dalton Müller Pessôa Filho.

**Methodology:** David Michel Oliveira, Mara Lucia Marques, Maria Claudia Bernardes Spexoto, Giovanna Benjamin Togashi, Danilo Alexandre Massini, Dalton Müller Pessôa Filho.

**Project administration:** David Michel Oliveira, Daniel dos Santos, Danilo Alexandre Massini, Dalton Müller Pessôa Filho.

**Resources:** David Michel Oliveira, Mara Lucia Marques, Daniel dos Santos, Maria Claudia Bernardes Spexoto, Dalton Müller Pessôa Filho.

**Software:** Mara Lucia Marques, Maria Claudia Bernardes Spexoto, Danilo Alexandre Massini.

**Supervision:** Mara Lucia Marques, Maria Claudia Bernardes Spexoto, Danilo Alexandre Massini, Dalton Müller Pessôa Filho.

**Validation:** Mara Lucia Marques, Daniel dos Santos, Maria Claudia Bernardes Spexoto, Giovanna Benjamin Togashi, Danilo Alexandre Massini, Dalton Müller Pessôa Filho.

**Visualization:** Mara Lucia Marques, Maria Claudia Bernardes Spexoto, Giovanna Benjamin Togashi, Danilo Alexandre Massini, Dalton Müller Pessôa Filho.

**Writing – original draft:** David Michel Oliveira, Mara Lucia Marques, Daniel dos Santos, Maria Claudia Bernardes Spexoto, Giovanna Benjamin Togashi, Dalton Müller Pessôa Filho.

**Writing – review & editing:** David Michel Oliveira, Giovanna Benjamin Togashi, Danilo Alexandre Massini, Dalton Müller Pessôa Filho.

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
