## [Decision Letter · Decision Letter 0]

11 Dec 2019

PONE-D-19-29357

Spatial index relating urban environment to health lifestyle and obesity risk in men and women from different age groups

PLOS ONE

Dear PhD Pessôa Filho,

Thank you for submitting your manuscript to PLOS ONE. After careful consideration, we feel that it has merit but does not fully meet PLOS ONE’s publication criteria as it currently stands. Therefore, we invite you to submit a revised version of the manuscript that addresses the points raised during the review process.

We would appreciate receiving your revised manuscript by Jan 25 2020 11:59PM. To enhance the reproducibility of your results, we recommend that if applicable you deposit your laboratory protocols in protocols.io, where a protocol can be assigned its own identifier (DOI) such that it can be cited independently in the future. For instructions see: http://journals.plos.org/plosone/s/submission-guidelines#loc-laboratory-protocols

We look forward to receiving your revised manuscript.

Kind regards,

Robert Siegel

Academic Editor

PLOS ONE

Journal Requirements:

2. Please include additional information regarding the survey or questionnaire used in the study and ensure that you have provided sufficient details that others could replicate the analyses. For instance, if you developed a questionnaire as part of this study and it is not under a copyright more restrictive than CC-BY, please include a copy, in both the original language and English, as Supporting Information. In addition, please provide any details of pre-testing of this questionnaire - i.e. how many participants were involved and from where were they recruited.

3. Please be wary of causal statements in your manuscript, including "the adoption of healthy life habits (practicing regular PA and avoiding smoking and excessive alcohol consumption) have a protective effect". Such statements cannot be supporting following a study of this design.

4. We note that Figure 1 in your submission contains satellite images which may be copyrighted. All PLOS content is published under the Creative Commons Attribution License (CC BY 4.0), which means that the manuscript, images, and Supporting Information files will be freely available online, and any third party is permitted to access, download, copy, distribute, and use these materials in any way, even commercially, with proper attribution. For these reasons, we cannot publish previously copyrighted maps or satellite images created using proprietary data, such as Google software (Google Maps, Street View, and Earth). For more information, see our copyright guidelines: http://journals.plos.org/plosone/s/licenses-and-copyright.

1.    You may seek permission from the original copyright holder of Figure(s) [#] to publish the content specifically under the CC BY 4.0 license. 

5.  Please ensure that you refer to Figure 1 in your text as, if accepted, production will need this reference to link the reader to the figure.

Additional Editor Comments (if provided):

Overall a very interesting study. An overall and simple explanation of the statistical approach would be helpful to many readers.

Reviewers' comments:

Reviewer's Responses to Questions

**Comments to the Author**

1. Is the manuscript technically sound, and do the data support the conclusions?

Reviewer #1: Yes

Reviewer #2: Yes

2. Has the statistical analysis been performed appropriately and rigorously? 

Reviewer #1: Yes

Reviewer #2: Yes

3. Have the authors made all data underlying the findings in their manuscript fully available?

Reviewer #1: Yes

Reviewer #2: Yes

4. Is the manuscript presented in an intelligible fashion and written in standard English?

Reviewer #1: Yes

Reviewer #2: Yes

5. Review Comments to the Author

Reviewer #1: This is an interesting paper as it seeks to develop a quantification tool to assess environmental factors that may contribute to activity levels and health lifestyle. This was a concept explored by the Australian group Leslie E et al., Health and Place 2007 but has never really been developed. The tool used four main indicators i.e. population density, road connectivity, area of green spaces and land incline in the calculation of the spatial index.

Relevant ethical approval appears to have been obtained for the study.

It is not clear how recruitment to the study was decided however to some degree it appears a biased population was obtained, as the study only sampled those who were outdoors using park or outdoor spaces. The study results may have been more powerful if this was compared with a control group of people approached within office spaces or homes within those areas. Alternatively comparing this population against a similar cohort from Goiania to verify results.

The paper is well written and statistical methods appear robust.

Further description is required regarding how the actual auxological/anthropometric measurements were taken (e.g. number of observers at any one time) and how standardisation was preserved to prevent significant inter or intraobserver error. Some increased clarity on this needs to be included.

Increasing BMI in the male group seems to correlate better with abdominal circumference than in the female group. This may have confounded the significance in result obtained in the two gender groups. A better measure may have been to not just look at BMI but using a portable bioimpedance scale to measure percentage fat versus muscle mass.

Although brief assessment of health risks was made in the study population such as smoking, dietary control and alcohol consumption, it is a shame that brief assessment of actual metabolic comorbidities such as heart disease and diabetes was not done and included in the analyses.

The discussion did effectively identify some of the limitations of the study, particularly to address the possible effect that the limited level of higher-intensity cardiovascular activity (running) was in the group and may have been contributory to the lack of statistical significance in the results.

When looking at this study along with other systematic reviews on effects of urban environments on health outcomes, it is clear that more confounders needs to be included in the measures such as actual equipment available for activity, indoor exercise spaces particularly in urban areas and population demographic that may be affected by ethnicity and other genetic factors.

Nevertheless, this paper is original in its approach and will almost certainly act as a study that will generate further research in the area.

Reviewer #2: Very interesting paper. Having an urban environment that promotes exercise is important and encourages exercise. However I was not surprised that there was no effect on BMI given the little impact that exercise plays in weight loss and changes in BMI. Especially when the main mode of activity is walking and given how little calories it burns when compared to other more intense exercise. Many studes show eating habits and genetic predispostion play a much larger role in weight loss and changes in BMI.

It would have been nice to look at body compostion as well to see if the higher BMI's were becasue of excess body fat or higher muscle mass in the more active population.

I found the statistical analysis very confusing but the paper was overall pretty easy to read and understand.

6. PLOS authors have the option to publish the peer review history of their article (what does this mean?). If published, this will include your full peer review and any attached files.

Reviewer #1: Yes: Mars Skae

Reviewer #2: No

---

## [Author Response · Author response to Decision Letter 0]

14 Feb 2020

The authors thank the reviewers for the interesting comments, which will improve the quality of the manuscript. All revised comments are shown in the document Response_to_Reviewers_PONE_D-19-29357.

---

## [Editor Report · Decision Letter 1]

19 Feb 2020

Spatial index relating urban environment to health lifestyle and obesity risk in men and women from different age groups

PONE-D-19-29357R1

Dear Dr. Pessôa Filho,

We are pleased to inform you that your manuscript has been judged scientifically suitable for publication and will be formally accepted for publication once it complies with all outstanding technical requirements.

With kind regards,

Robert Siegel

Academic Editor

PLOS ONE

Additional Editor Comments (optional):

The authors are to be commended for addressing all the reviewer concerns and incorporating all the requested changes.

The manuscript is ready for publication.
---

## [Editor Report · Acceptance letter]

26 Feb 2020

PONE-D-19-29357R1 

Spatial index relating urban environment to health lifestyle and obesity risk in men and women from different age groups 

Dear Dr. Pessôa Filho:

I am pleased to inform you that your manuscript has been deemed suitable for publication in PLOS ONE. Congratulations! Your manuscript is now with our production department. 

With kind regards,

on behalf of

Dr. Robert Siegel 

Academic Editor

PLOS ONE